# Protein Biomarkers Enable Sensitive and Specific Cervical Intraepithelial Neoplasia (CIN) II/III+ Detection: One Step Closer to Universal Cervical Cancer Screening

**DOI:** 10.3390/cancers17111763

**Published:** 2025-05-24

**Authors:** Samrin F. Habbani, Sayeh Dowlatshahi, Nathanael Lichti, Meaghan Broman, Lucy Tecle, Scott Bolton, Lisa Flowers, Rafael Guerrero-Preston, Jacqueline C. Linnes, Sulma I. Mohammed

**Affiliations:** 1Department of Comparative Pathobiology, Purdue University, West Lafayette, IN 47907, USA; shabbani@purdue.edu (S.F.H.); mbroman@purdue.edu (M.B.); 2Weldon School of Biomedical Engineering, Purdue University, West Lafayette, IN 47907, USA; sjalalid@purdue.edu (S.D.); ltecle@purdue.edu (L.T.); boltons@purdue.edu (S.B.); jlinnes@purdue.edu (J.C.L.); 3Bindley Bioscience Center, Purdue University, West Lafayette, IN 47907, USA; nlichti@purdue.edu; 4Department of Obstetrics and Gynecology, Emory University School of Medicine, Atlanta, GA 30322, USA; lflowe2@emory.edu; 5LifeGene Biomarks, Toa Baja 00949, Puerto Rico; rguerrero@lifegenedna.com; 6Department of Small Animal Clinical Sciences, Cancer Control and Population Sciences, University of Florida Health Cancer Center, Gainesville, FL 32610, USA

**Keywords:** protein biomarkers, cervical precancerous lesions, cervical intraepithelial neoplasia, high-grade squamous intraepithelial lesions, cervical cancer, early detection, low- and middle-income countries

## Abstract

Cervical cancer (CC) is the fourth most common cancer among women globally and disproportionately affects those in low- and middle-income countries (LMICs), in which 84% of CC cases and 87–90% of CC deaths occur each year. Current screening methods are expensive, time-consuming, and often unavailable in resource-limited settings. This study validates critical protein biomarkers, including topoisomerase II alpha (TOP2A), minichromosome maintenance complex component 2 (MCM2), valosin-containing protein (VCP), and cyclin-dependent kinase inhibitor 2A (p16INK4a), for point-of-care CC screening. By optimizing a simple and efficient protein extraction protocol, we further demonstrate the potential application of these biomarkers to detect precancerous lesions and distinguish CC subtypes. Our results facilitate the development of low-cost, rapid, and accurate screening assays, with the potential to transform early-stage CC diagnosis and treatment in LMICs.

## 1. Introduction

Cervical cancer (CC) remains a significant global health challenge for women, ranking as the fourth most common cancer worldwide and contributing to a substantial number of cancer-related deaths annually [1,2]. A stark disparity in CC incidence exists between high-income and low- and middle-income countries (LMICs), with 87–90% of annual deaths occurring in these regions [1]. This unequal burden is primarily due to disparities in access to human papillomavirus (HPV) vaccination and effective CC screening as primary and secondary prevention procedures, respectively [1,2,3].

The progression of CC follows defined stages, starting with persistent high-risk HPV infection, followed by dysplasia, and, ultimately, invasive carcinoma (Appendix A). Benign changes in cervical epithelial cells are classified as reactive cellular changes (RCC), which fall under the category of negative for intraepithelial lesion or malignancy (NILM) [4,5]. Atypical squamous cells of undetermined significance (ASC-US) represent the most common abnormal finding in cervical cytology. This classification is applied to Papanicolaou (Pap) smear results that display abnormal cytologic changes suggestive of a squamous intraepithelial lesion (SIL), but do not fulfill the necessary criteria for a definitive SIL diagnosis [5,6,7]. Hence, ASC-US is seen as an indeterminate result, often necessitating additional testing to evaluate the risk of underlying cervical intraepithelial neoplasia (CIN) development [8]. Cervical precancerous lesions, though not malignant, have the potential to progress to cancer if left undetected. Low-grade squamous intraepithelial lesions (LSILs) may regress spontaneously, whereas high-grade squamous intraepithelial lesions (HSILs) are more likely to progress to invasive carcinoma [9,10,11,12]. As dysplasia progresses to CC, the malignancy is highly heterogeneous and is characterized by various histological subtypes, with squamous cell carcinoma (SCC) accounting for 75–90% of cases, followed by adenocarcinoma (ADC) [13]. Glassy cell carcinoma (GCC), a rare, aggressive CC subtype, is characterized by rapid growth and a high likelihood of early metastasis [14].

Owing to the World Health Organization’s call for CC elimination, there has been a significant improvement in CC prevention through HPV vaccination in LMICs recently [15,16]. Nonetheless, more efficient vaccination programs need to be in place in these regions to achieve the >80% vaccination rates as in developed countries [16]. Moreover, there has been extensive research on the post-infection application of the HPV vaccine as adjuvant therapy, preventing the progression of HSILs to invasive CC [17].

Despite significant progress in HPV vaccination, current CC screening methods, such as cytology-based Pap smear testing and HPV DNA testing, face limitations, including low sensitivity and/or specificity, high costs, and reliance on skilled operators and infrastructure [12,18,19,20]. These challenges highlight the urgent need for novel, accurate, cost-effective screening techniques, particularly for LMICs. Developing histologic and genetic screening methods has been challenging due to the variable microenvironments of cervical precancer and cancer lesions, as well as the presence of some HPV-independent cases [21,22]. As an alternative, novel detection strategies based on protein biomarkers may allow for sensitive, specific point-of-care (POC) tests while minimizing the need for expensive equipment or extensive training.

As the first phase of developing a point-of-care (POC) CC screening test, this study evaluates the potential of four biomarkers—topoisomerase II alpha (TOP2A), minichromosome maintenance complex component 2 (MCM2), valosin-containing protein (VCP), and cyclin-dependent kinase inhibitor 2A (CDKN2A or p16INK4a)—to detect cervical precancerous lesions [23]. These biomarkers were selected based on their known dysregulation following oncogenic mutations in the cervix. TOP2A is a DNA replication enzyme overexpressed in CINs and invasive CC [24,25]. MCM2, a cell cycle regulatory protein, is highly expressed in cervical and other cancers [22,25]. VCP plays a role in endoplasmic reticulum-associated degradation and shows elevated expression in multiple cancer types, including that of the cervix [2]. Lastly, p16INK4a is a tumor suppressor protein strongly associated with cervical dysplasia and carcinoma [22,25,26]. To perform efficient biomarker validation, we first evaluated various cell isolation and lysis techniques and then measured the four proteins in cultured cervical cells and clinical cervical swabs using enzyme-linked immunosorbent assays (ELISAs). We also performed immunohistochemistry (IHC) on tissue microarrays (TMAs) to quantify the biomarkers in cervical tissues. By focusing on protein markers, we aim to address critical gaps in CC screening and prevention in resource-limited settings, facilitating our next step, which is the development of a user-friendly cost-effective screening test that requires minimal training and infrastructure.

## 2. Materials and Methods

### 2.1. Cell Culture

The following CC cell lines were used: HeLa (#CCL-2™, RRID: CVCL_0030Ca Ski (#CRL-1550™, RRID: CVCL_1100), HT-3 (#HTB-32™, RRID: CVCL_1293), and C-33 A (#HTB-31™, RRID: CVCL_1094). Primary cervical epithelial cells (PCS cells, #PCS-480-011™) were also employed as a negative control. All these cells were well characterized and purchased from the American Type Culture Collection (ATCC, Manassas, VA, USA), and they have tested negative for mycoplasma contamination. HeLa and HPV-negative C-33 A cells were grown separately in Eagle’s Minimum Essential Medium (#30-2003, ATCC). Ca Ski cells were cultured in RPMI-1640 Medium (#30-2001, ATCC). HT-3 cells were cultured in McCoy’s 5A Medium (#30-2007, ATCC). The cell line media were supplemented with 10% fetal bovine serum (#30-2020, ATCC). PCS cells were cultured in Cervical Epithelial Cell Basal Medium (#PCS-480-032, ATCC), supplemented with its Growth Kit (#PCS-480-042, ATCC).

### 2.2. Comparison of Buffer Efficiency in Cell Lysis and Protein Recovery from Cultured Cervical Cells

To identify the buffer with optimal lysis efficiency and protein recovery, two common lysis buffers were tested on the four CC cell lines (HeLa, Ca Ski, HT-3, and C-33 A) and PCS cells. The lysis buffers were PARIS™ cell disruption buffer (#AM1921, Thermo Fisher Scientific, Waltham, MA, USA) and radioimmunoprecipitation assay (RIPA) buffer (10 mM Tris-HCl, 1% Triton X-100, 0.1% sodium dodecyl sulfate (SDS), 0.1% sodium deoxycholate, 140 mM NaCl, 1 mM EDTA, and 0.5 mM EGTA). Cultured cells were harvested at 85% confluency, counted using Countess™ 3 FL Automated Cell Counter (Thermo Fisher Scientific), washed in ice-cold phosphate-buffered saline (#AM9625, Thermo Fisher Scientific), and pelleted by centrifugation. The pellets were lysed with the designated buffers and homogenized. The total protein concentration of the lysate samples was measured using Pierce™ bicinchoninic acid (BCA) protein assay (#23227, Thermo Fisher Scientific).

### 2.3. Protein Extraction from Clinical Cervical Swab Samples

Cervical swabs for this study were obtained from two collaborators at different institutions. LifeGene Biomarks provided swabs collected from women undergoing cytology testing at CLIA-certified US laboratories under an Institutional Review Board (IRB) (IRB00231112, Johns Hopkins, Baltimore, MD, USA). Broom-type or Cytobrush/spatula devices were used to sample the patients. Specimens were rinsed into ThinPrep^®^ Pap Test vials with PreservCyt^®^ Solution (Hologic^®^, Marlborough, MA, USA) and underwent Pap smear testing and histopathological analysis. This set included swabs from normal, CIN II, and CIN III cases. The second set of cervical swab specimens was collected from African American women undergoing liquid-based Pap test screening at Grady Health System/Emory University in Atlanta, GA, USA, under IRB00112522. The patients were sampled using either the Medi-Pak cytology brush (Puritan Medical Products, Guilford, ME, USA) or the Cervex-Brush^®^ cytology broom (Puritan Medical Products), and the collected specimens were preserved in BD SurePath™ preservative fluid (BD Biosciences, Franklin Lakes, NJ, USA). This set included normal, RCC, ASC-US, LSIL, and HSIL cases that were randomly selected. De-identified samples were provided to Purdue University under a Material Transfer Agreement, and secondary analysis was performed on the specimens at Purdue University under IRB2022816.

Two cell isolation techniques were employed to extract cells from the clinical cervical swab specimens. The first approach used acetone precipitation to remove contaminants and concentrate cells directly from the preservative solution. The second approach utilized a filter cartridge to capture and isolate cells from the preservative solution. Upon cell extraction, the isolated cell pellets from both methods were lysed using either RIPA or PARIS™ lysis buffers. The resulting lysates’ total protein concentration was quantified using the BCA assay to determine the optimal cell isolation + protein extraction technique.

### 2.4. Biomarker Detection and Quantification Using ELISAs

Commercial ELISA kits were employed to quantify the four biomarkers and beta-actin, as the human positive control, in lysates of cultured cells and clinical swab samples. TOP2A (#EKF58744), MCM2 (#EKN47079), p16INK4a (#EKN52207), and beta-actin (#EKN49629) ELISA kits were purchased from Biomatik (Kitchener, ON, Canada). VCP (#CSB EL025813HU) was obtained from CUSABIO (Houston, TX, USA). All assays followed a sandwich-type ELISA format and were performed according to the manufacturer’s protocols.

### 2.5. Biomarker Detection Using ICC

Cytology slides were prepared by directly smearing cervical swab samples onto glass slides. The cells were fixed in acetone and stained using Hematek^®^ Stain Pak-Modified Wright’s Stain (Siemens Healthcare Diagnostics, Tarrytown, NY, USA). Subsequently, the slides were processed for ICC. Cells were permeabilized with 0.1% Triton X-100, blocked with 5% bovine serum albumin (BSA), and incubated overnight with anti-cytokeratin 19 (CK19—an epithelial cell marker) clone 4E8 (#NBP2-22116, Novus Biologicals, Centennial, CO, USA). Slides were then treated with the secondary antibody reagent, MACH 4 Universal HRP-Polymer (#50-829-12, Biocare Medical, Pacheco, CA, USA), and developed with 3,3′-diaminobenzidine (DAB, #BDB550880, BD Biosciences). Counterstaining was performed with Mayer’s hematoxylin (Richard-Allan Scientific, Kalamazoo, MI, USA). The slides were scanned with Motic EasyScanner, RRID:SCR_024855 (Motic Digital Pathology, Emeryville, CA, USA). The images were visualized using Motic VM 3.0 Digital Slide Assistant, version 1.0.7.61b, licensed by the State University of New York (Albany, NY, USA).

### 2.6. IHC on Cervical TMAs

The TMAs used in this study were obtained from TissueArray.Com (Derwood, MD, USA): CR486 (LSIL, HSIL, and SCC, 24 cases/48 cores), T104a (SCC, GCC, and ADC, 6 cases/24 cores), and CR1002 (SCC and ADC, 25 cases/100 cores). IHC was performed according to the Biocare Medical protocol. Initially, the slides containing the tissue sections were deparaffinized in xylene and then hydrated in graded ethanol concentrations. Endogenous peroxidase activity was quenched using Biocare Medical’s PeroxAbolish (#PXA969M). Heat-induced epitope retrieval was then performed by placing the slides in Biocare Medical’s Diva Decloaker antigen retrieval solution (#DV2004MX) at 95 °C in a pressure cooker. Subsequently, the slides were blocked with Biocare Medical’s Background Punisher (#BP974M) and incubated in primary antibodies. The antibodies included anti-human TOP2A clone 1E2 (#H00007153-M01, RRID: AB_607207, Novus Biologicals), anti-human MCM2 clone EPR4120 (#ab108935, RRID: AB_10859977, Abcam, Waltham, MA, USA), anti-human VCP/p97 clone EPR3307 (2) (#ab109240, RRID: AB_10862588, Abcam), and anti-human p16INK4a/CDKN2A clone 1D7D2A1 (#NBP245602, RRID: AB_3309744, Novus Biologicals). This was followed by a 15 min incubation in the mouse probe, a 10 min incubation in the HRP polymer, and a 5 min incubation in DAB. The slides were then immersed in Mayer’s hematoxylin. IHC sections were digitized using an Aperio Digital Pathology Slide Scanner (Leica Biosystems, Deer Park, IL, USA) and quantitatively analyzed using the Visiopharm Digital Slide Analysis Platform, version 2024.07.2.17285x64, RRID:SCR_021711 (Visiopharm, Hørsholm, Denmark). A board-certified pathologist conducted the histomorphologic confirmation and immunostain quantifications of the studied TMAs. For each sample, tumor and stromal regions were segmented, and only the tumor regions of non-fragmented cores were analyzed. The labeling intensity of tumor cell nuclei was quantified and categorized on a scale of +1 to +3. For each sample, the proportion of tumor nuclei positive for a specific biomarker to the total tumor nuclei was also calculated.

### 2.7. Statistical Analyses

The biomarker concentrations were measured for each cultured cell type and clinical swab’s cytology category using two technical ELISA replicates for each biological replicate. The technical replicates were averaged and internally normalized to the beta-actin concentration in the same sample before the analyses. The concentrations were then analyzed using Gamma-distributed log-link generalized linear mixed models (GLMMs) with fixed effects for protein, cell type, or cytology class, and their interactions, plus random effects for biological replicates within each cell type or cytology group. Contrasts were estimated on the log scale and then exponentiated to yield confidence intervals for fold change. In addition to the GLMM analysis, principal component analysis (PCA) was used to visualize the multivariate differences among samples and their relationships to the proteins. Analyses were conducted in R, version 4.4.1, (RRID: SCR_001905), using the lme4 and emmeans packages for GLMMs and post hoc comparisons [27,28].

In addition to the GLMM analysis, clinical swab specimens were classified into cytology categories using a multinomial generalized additive model (GAM) fitted to the beta-actin normalized concentrations of the target biomarkers. GAMs are semiparametric regression models that fit smooth, nonlinear functions of explanatory variables. In this case, thin-plate regression splines were applied to categorize the measurements into normal, RCC, ASC-US, LSIL, or HSIL. GAMs were implemented in the R package mgcv, version 1.9-1 [29]. Cases were assigned to the class with the highest predicted probability. To evaluate model performance, we analyzed the confusion matrix of actual versus predicted classes, calculated the balanced error rate (BER), and conducted a bootstrap cross-validation with 200 replications. In each replicate, 25% of cases in each class were reserved as a test set, and the model was fit to the remaining 75% of the data.

The IHC data were analyzed by calculating the total number of tumor nuclei and the proportion of positive nuclei (those scoring ≥ +1 for a given protein) for each case ID (duplicates were averaged when they occurred). A logit-link quasibinomial generalized linear model was used to estimate the proportion of tumor nuclei positive for each biomarker in tissues with LSIL, HSIL, SCC, or GCC diagnosis. The quasibinomial family was used to better accommodate the overdispersion observed in the data. Each sample was weighted by the total number of tumor nuclei seen in the core. Similar methods were used to compare SILs and SCCs G1–3.

We built a GAM classifier similar to that described for swab samples above to categorize the TMA specimens as LSIL, HSIL, or SCC using the IHC data. Prior to model fitting, proportions were linearized through a logit transformation. Due to the small number of LSIL samples (*n* = 9), one-third of the data were reserved for cross-validation testing rather than 25%. Additionally, GCC samples were excluded from the analysis, given their limited availability.

GraphPad Prism 10.2.1 (RRID: SCR_002798) was employed to perform other statistical tests, including one-way ANOVA and Welch’s *t*-test, as well as subsequent multiple comparisons tests, e.g., Tukey’s or Dunnett’s, to assess the measurements’ *p*-values and their statistical significance (*p* < 0.05, with 95% confidence interval).

## 3. Results

### 3.1. Comparative Analysis of Protein Yield from CC Cell Lines and PCS Cells Using Different Lysis Buffers

A reliable protein-based POC test requires an efficient protein extraction method. This study assessed two lysis buffers, PARIS™ and RIPA, across four CC cell lines and PCS cells with an equal cell count of 4 × 10^6^. RIPA demonstrated superior performance compared to PARIS™ for four of the five cell types, and, therefore, it was selected for lysing cervical cells from cell cultures and swab specimens (Figure 1).

### 3.2. Biomarker Expression in Cultured Cervical Cells

Protein expression levels of the four biomarkers TOP2A, MCM2, VCP, and p16INK4a, were quantified using ELISAs in the four cervical cell lines, HeLa, Ca Ski, HT-3, and C-33 A, as well as PCS cells (4 × 10^6^ cells/sample). TOP2A and MCM2 were present in higher amounts per cell in PCS lysates compared to cancer cell lines. In contrast, VCP and p16INK4a levels were significantly elevated in HeLa relative to PCS cells (Appendix A).

To compensate for unequal protein loading, beta-actin levels were also measured in each sample (Appendix A) and were used to normalize the biomarker concentrations. As shown in Figure 2, the analysis revealed that VCP and p16INK4a were overexpressed in all cancer cell lines compared to PCS cells. MCM2 was overexpressed in HT-3 and HeLa but not in Ca Ski or C-33 A relative to PCS cells. TOP2A was underexpressed in Ca Ski compared to other cancer cell lines. Expression patterns highlighted similarities between HT-3 and HeLa, as well as between Ca Ski and C-33 A (Figure 2A). PCA demonstrated a distinct separation between PCS cells and cancer cell lines, driven by elevated VCP and p16INK4a in cancer cells. PCA further distinguished HeLa from C-33 A and Ca Ski from HT-3 based on varying levels of MCM2, VCP, and p16INK4a (Figure 2B).

### 3.3. Optimization and Clinical Translation of Total Protein Extraction from Clinical Cervical Swab Samples

To translate this work to a clinical setting, the protein extraction capabilities of RIPA and PARIS™ were also compared in clinical cervical swabs. Modified Wright’s staining on samples from normal, LSIL, and HSIL cases revealed well-preserved cervical squamous cell morphology across all cytology categories (Figure 3A). Additionally, ICC staining of these samples showed positive expression for CK19, indicating the presence of epithelial cells (Figure 3B). Regarding cell isolation and lysis, cervical cells stored in BD SurePath™ media were extracted more efficiently with higher protein recovery using acetone precipitation than filter-based isolation (Figure 3C). Moreover, combining acetone precipitation with RIPA buffer consistently produced higher protein concentrations than combining it with PARIS™ lysis buffer (Figure 3D). Lastly, analysis of a larger sample group confirmed the efficacy of the acetone precipitation (for cell isolation) + RIPA (for cell lysis) method across different swab categories (Figure 3E).

### 3.4. Biomarker Expression in Clinical Cervical Swab Samples

The expression of the four biomarkers was quantified in cervical swab lysates using ELISAs to assess their trends across different CIN or cytology categories. A pilot study examined biomarker expressions in swab lysates from normal (*n* = 4), CIN II (*n* = 8), and CIN III (*n* = 8) samples (Figure 4A). MCM2 and p16INK4a ratios were elevated in CIN III samples compared to normal specimens.

For the samples with different cytology categories, the HSIL swabs had significantly higher VCP ratios compared to the LSIL, ASC-US, and normal samples. p16INK4a ratios were higher in HSIL, LSIL, and ASC-US specimens compared to normal samples. Though elevated in HSIL lysates, TOP2A and MCM2 ratios were not statistically significantly higher than those of other cytology categories (Figure 4B). A PCA on the protein markers showed substantial overlap among the cytology groups. While RCC samples demonstrated stronger overlap with normal specimens, HSIL and, to a lesser extent, LSIL samples differed noticeably from the normal category. ASC-US appeared to be the most dysregulated. Based on this analysis, VCP and p16INK4a were overexpressed in most LSIL and HSIL samples, while TOP2A and MCM2 were overexpressed in HSIL specimens only. In some cases, TOP2A was underexpressed in LSIL samples (Figure 4C). These findings underscore the potential of MCM2, VCP, and p16INK4a in distinguishing between precancerous lesions (LSIL and/or HSIL) and healthy or benign samples (normal or RCC).

#### Evaluating the Classification Potential of the Four Biomarkers in Lysates of Clinical Cervical Swab Specimens

A classification model was developed for the clinical cervical swab samples quantified in Figure 4B. The model excluded MCM2 and TOP2A because their beta-actin-normalized values showed very few differences among the categories. Furthermore, because ASC-US represents an uncertain classification that likely includes LSIL and HSIL cases, we fitted the model both without and with ASC-US data. In addition, because the model had difficulty distinguishing the RCC and normal categories, and both of these are deemed clinically negative, we discussed them as a single NILM category.

Predictions and receiver operating characteristic (ROC) curves for the classifier fitted to the above-mentioned training data are shown in Figure 5. Without the ASC-US samples, the model achieved perfect in-sample discrimination among the NILM, LSIL, and HSIL categories (BER = 0). Overall, HSIL findings were accurately predicted by a relatively high ratio of VCP to beta-actin, while LSIL was predicted by a high ratio of p16INK4a to beta-actin in the absence of elevated VCP. Additionally, an inspection of the confusion matrix revealed that 97.3% of misclassified HSIL cases in the test set were assigned to LSIL. Misidentified LSIL cases tended to be classified as normal (53.2%) or HSIL (39.3%). 100% of misassigned RCC cases were identified as normal, and 83.5% of misidentified normal cases were assigned to RCC (Figure 5A). Cross-validation-based ROC curves are shown in Figure 5B. At their optimal points, the sensitivity and specificity of each category versus all other groups were as follows: HSIL (95.6% and 90.4%), LSIL (89.2% and 87.6%), RCC (78% and 81.5%), and normal (95.6% and 60.7%).

The inclusion of ASC-US as a possible outcome reduced the model’s overall accuracy (BER = 45.3%), with the majority of normal and LSIL samples misclassified as ASC-US (Figure 5C). In cross-validation, the model achieved acceptable optimal sensitivity and specificity for HSIL (90.8% and 90.2%, respectively); however, these metrics were lower for the other cytology categories: LSIL (75.3% and 75.1%), ASC-US (75.1% and 58.1%), RCC (85.7% and 79.1%), and normal (84.0% and 61.0%) (Figure 5D).

### 3.5. Biomarker Expression in Cervical Precancer and Cancer Tissues

#### 3.5.1. Biomarker Expression Trends in Tissues of Cervical SILs and CC Subtypes

The proportion of tumor nuclei stained positively for each biomarker was analyzed in tissues of cervical SILs (LSIL and HSIL) and two CC subtypes (GCC and SCC), as illustrated in Figure 6. A significant increase in biomarker expression was observed with disease progression, with LSIL tissues displaying the lowest expression levels and carcinoma cases (GCC or SCC) showing the highest (Figure 6A–D). HSIL consistently demonstrated higher expression than LSIL for all four biomarkers; however, the overexpression of TOP2A in HSIL tissues did not achieve statistical significance (Figure 6A). GCC tissues exhibited variable biomarker expression patterns, with MCM2 levels being statistically significantly higher than those of LSIL arrays (Figure 6B). Of note, the one ADC case (four cores) was excluded from the main analysis; however, the expression of all biomarkers in these tissues was analyzed (Appendix A). Notably, TOP2A, VCP, and p16INK4a showed strong staining compared to MCM2.

#### 3.5.2. Biomarker Expression Trends in Tissues of Cervical SILs and Different Histopathology Grades of SCC

The analysis of LSIL and HSIL tissues, as well as SCC tumor samples of grades G1–3, revealed distinct expression patterns for the four proteins (TOP2A, MCM2, VCP, and p16INK4a) (Figure 7). All the biomarkers demonstrated statistically significant differences between LSIL and G2 SCC. MCM2 and p16INK4a also exhibited significant differences between LSIL and G3 SCC. TOP2A had statistically significantly higher levels in G2 and G3 SCC than in HSIL. Collectively, the observed trends supported the progressive overexpression of the biomarkers in HSIL and SCC tissues (Figure 7A–D).

#### 3.5.3. Evaluating the Classification Potential of the Four Biomarkers in Cervical Tissues

Based on Figure 6B,C, the proportion of tumor cells expressing MCM2 and VCP was identified as the primary predictive feature in the classification model. The inclusion of TOP2A or p16INK4a did not enhance the model’s predictive performance within this dataset. The classifier predictions and corresponding histopathological diagnoses are presented in Figure 8A. The model achieved a BER of 0.07% for in-sample classification. All LSIL, 11/12 HSIL cases, and 25/28 SCC cases were correctly classified. In general, HSIL and SCC were characterized by progressively increasing proportions of MCM2- and VCP-positive nuclei.

In cross-validation, the model achieved a sensitivity and specificity of 99.3% and 92.5% for LSIL, 92.4% and 74.7% for HSIL, and 85.4% and 93.1% for SCC (Figure 8B). Misclassifications of both LSIL and SCC were always as HSIL, whereas misidentified HSIL cases were split, with 28% predicted as LSIL and the remainder as SCC.

## 4. Discussion

This study quantified and validated four cervical precancer and cancer biomarkers—TOP2A, MCM2, VCP, and p16INK4a—in lysates of cultured HeLa, Ca Ski, HT-3, and C-33 A cancer cell lines, as well as normal PCS cells, clinical cervical swab specimens, and cervical tissues. The aim was to better understand their expression levels quantitatively and to assess their potential application as targets in a liquid biopsy-based POC screening test. Most CC screening techniques are based on microscopic cell analysis or HPV DNA detection instead of analyzing proteins in the swab samples. This may be due to the challenges associated with extracting proteins from complex mucous mixtures [30]. By focusing on protein analysis, this study investigated potential biomarkers to improve current screening techniques and facilitate early detection, particularly in LMICs. To the best of our knowledge, this work is the first of its kind, as previous studies on these CC-associated biomarkers solely validated their overexpression in cervical tissues and cytology samples qualitatively [2,22,31].

A unified protocol to isolate cells, extract proteins, and measure the target protein markers was optimized and validated for cervical swab samples collected into transport media. The acetone precipitation technique was shown to be highly effective in separating cells from the archived swabs. This aligned with previous studies that highlighted this method’s effectiveness in removing contaminants and preserving the cell structure for further protein-related analyses [32,33].

In this study, achieving adequately high protein yields was key to guaranteeing the reproducible testing of each sample for all biomarkers. All four target proteins in this study are localized to the nucleus, particularly the nucleoplasm [34,35,36]. RIPA was able to solubilize them with an overall higher lysis efficiency than PARIS™ and was therefore chosen for protein extraction in this work. The findings matched with the previous literature reporting RIPA as the gold standard for cell lysis, offering rapid and efficient solubilization of membrane, cytoplasmic, and nuclear proteins [37].

Despite using RIPA buffer, preparing the lysates from 4 × 10^6^ cells, and utilizing beta-actin normalization, we observed different expression levels for each protein in each cell type. These differences aligned with the previous literature investigating biomarker levels among different cervical cell types and tissue lysates. A study on TOP2A revealed varying expression levels in six cervical cancer cell lines; SiHa displayed the highest levels, while HeLa showed the lowest [38]. Additionally, genomic studies, including mRNA analysis, demonstrated higher MCM2 levels in four cervical cancer cell lines compared to a normal cervical epithelial cell line [39]. Moreover, VCP had higher levels in tumor tissue lysates than lysates of normal tissues [2]. Finally, variations in p16INK4a expression were studied in three cervical cancer cell lines; however, this work used this marker to investigate retinoblastoma gene-mediated cell cycle arrest [40].

The expression variations observed in our research could have been due to differences in cell doubling times and the presence of high-risk HPV infection. Regarding the duplication times, the PCS cells have the lowest doubling time of 4–6 days, followed by Ca Ski (3.2 days), HT-3 (2.48 days), C-33 A (1.36 days), and HeLa (1.3 days) [41,42]. Studies have shown that, during the cell cycle, both TOP2A and MCM2 are overexpressed in the dysregulated S phase [25]. Additionally, flow cytometry has revealed that VCP and p16INK4a levels peak during the G0/G1 and late G1 phases, respectively [43,44]. In our experiments, the harvest time was similar to the time of subculture, i.e., the late log phase before full confluency or late G2/M phase [45].

Another factor influencing biomarker overexpression is persistent HPV infection. The HPV E6/E7 oncogenes and oncoproteins mediate the overexpression of the four target biomarkers [2,21,25]. As a result, the HPV-negative C-33 A cell line, which was used as a model for HPV-independent CCs, demonstrated lower overexpression for all the protein markers.

Regarding the PCS cells, besides having been isolated directly from the cervix, there is no other information available on how they were collected, handled, and processed [42]. Therefore, it is not surprising that, in our study, the protein expression for these cells was considerably different from that of the NILM cervical swabs.

Only a few studies have investigated these biomarkers in clinical cervical swabs. Earlier research examining mRNA biomarkers in liquid-based cytology (LBC) specimens underscored the potential of TOP2A, along with p16INK4a and Ki-67, for detecting HSIL [46]. Additionally, Del Moral-Hernández et al. found that TOP2A, MCM2, p16INK4a, and cyclin E1 were overexpressed in high-risk HPV-positive LBC samples, with levels increasing with lesion severity [22]. Moreover, there are a few tests available on the market that target some of these proteins: the ProEx™ C cytology test targets TOP2A and MCM2 proteins and detects HPV-induced aberrant S-phase [47]. Another test is CINtec^®^ PLUS Cytology, which simultaneously identifies p16INK4a and Ki-67 in HPV-positive cytology specimens. The co-expression of p16INK4a and Ki-67 is highly associated with HPV infection’s oncogenic transformation [25,48]. Altogether, these studies and commercial tests rely solely on microscopy-dependent cytological diagnoses, not including any quantitative techniques. In contrast, the work presented in this article stands out as it employed a comprehensive quantitative approach utilizing ELISAs. This research provides a groundbreaking assessment of the four proteins in clinical cervical swab samples across a broad spectrum of cytology classifications. The results underscored the potential of MCM2 and p16INK4a as biomarkers indicative of CIN III lesions (Figure 4A). Additionally, these findings highlighted the multiplexing potential of all four markers—TOP2A, MCM2, VCP, and p16INK4a—for differentiating HSIL cases from the less severe categories (Figure 4B,C). As demonstrated in Figure 5, we evaluated the classification potential of beta-actin-normalized VCP and p16INK4a in identifying HSILs. The classifier achieved a sensitivity and specificity of 95.6% and 90.4%, respectively, which is a significant advancement over gold-standard Pap smear testing that has a 51.5% sensitivity and 83.4% specificity for CIN II+ biopsy cut-offs [49,50].

This research also utilized a TMA approach with various histopathological diagnoses and grades to validate protein expression profiles (Figure 6 and Figure 7). The analysis confirmed positive staining of all target biomarkers in both HSIL and cancerous tissues, including SCC G1-3, thus aligning with previous IHC-based research and confirming their potential role in CC progression [2,22,25,51]. Of note, we also developed a classification model based on the proportion of MCM2- and VCP-positive nuclei that distinguished SCC tissues from LSILs and HSILs with 85.4% sensitivity and 93.1% specificity. This classification strategy surpassed the reported IHC-based detection of CIN II/III+ through VCP staining, which had a sensitivity and specificity of 93.0% and 87.0%, respectively [2]. Though the current commercial IHC tests do not target these two proteins, CINtec^®^ Histology, which detects p16INK4a, has shown a sensitivity and specificity of 87.0% and 88.0%, respectively [52].

Despite the promising findings of this study, there were several challenges. First, while these biomarkers have shown significant overexpression in dysplastic cervical tissues, their levels in cervical swabs were quite low. The relatively small sample size might have contributed to this discrepancy. Furthermore, this inconsistency might have also arisen because tissue specimens comprise all the layers of the cervical epithelium, while swab samples solely contain surface epithelial cells [25]. Additionally, sample collection and handling variations can introduce biases and affect biomarker stability [53]. Lastly, another limitation was the rarity of swab samples with high-grade cytology in routine screening, impacting the generalizability of the findings. Future studies should adopt a universal sample collection and handling protocol suitable for clinics in different settings. Moreover, further follow-up on ASC-US cases and a larger sample size for HSIL cases should be included to better validate the clinical applicability of these biomarkers.

## 5. Conclusions

This study highlighted the crucial role of the four biomarkers in cervical disease progression. It also underscored the heterogeneous protein expression in LBC samples, demonstrating the need to simultaneously detect these biomarkers for accurate CC screening. The next steps include optimizing a standardized cervical sample collection and processing protocol and translating the quantitative biomarker results into a POC multiplexed test, offering an affordable and scalable solution for improving global CC prevention. Additionally, given the expression variations, supervised machine learning will be employed for accurate results interpretation.

## Figures and Tables

**Figure 1 cancers-17-01763-f001:**
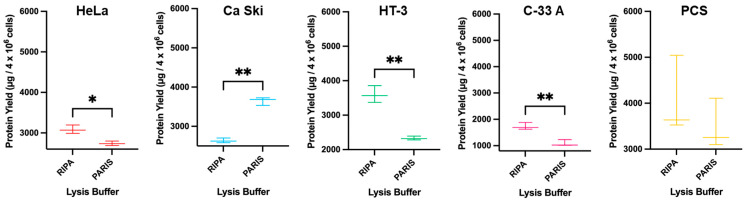
Protein yield across four CC cell lines and PCS cells using RIPA and PARIS™ lysis buffers. Total protein quantification was performed using BCA assay on 4 × 10^6^ cells of each CC cell line (HeLa, Ca Ski, HT-3, and C-33 A) and PCS cells lysed with RIPA or PARIS™. (Paired *t*-test: *: *p* < 0.05, **: *p* < 0.01, *n* = 3. Boxplots represent the median (bar) and range (whiskers) of the data.)

**Figure 2 cancers-17-01763-f002:**
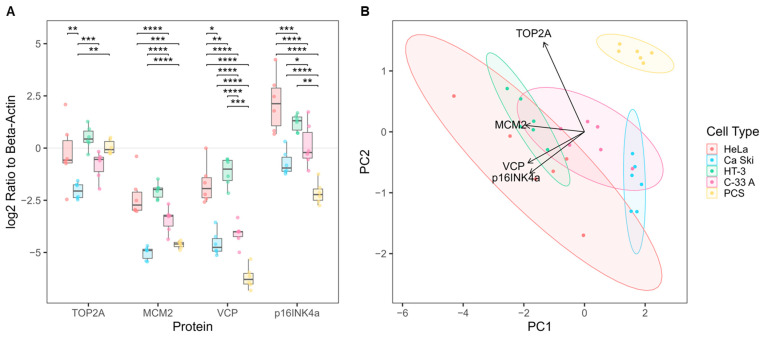
Biomarker expression reported as ratios to beta-actin in lysates of CC cell lines and PCS cells. (**A**) ELISA measurements of TOP2A, MCM2, VCP, and p16INK4a normalized to beta-actin in lysates of HeLa, Ca Ski, HT-3, and C-33 A cancer cell lines, and PCS cells. Statistical significance was assessed using a Gamma-distributed log-link GLMM, with fixed effects for proteins, cell types, and their interactions, and random effects for biological replicates (*n* = 3). (Holm-corrected pairwise *t*-test: *: *p* < 0.05, **: *p* < 0.01, ***: *p* < 0.001, ****: *p* < 0.0001. Boxplots represent the median (bar), interquartile range (IQR, box), and 1.5 × IQR (whiskers); the points correspond to individual data points.) (**B**) PCA of biomarker concentrations across cultured CC cell lines and PCS cells.

**Figure 3 cancers-17-01763-f003:**
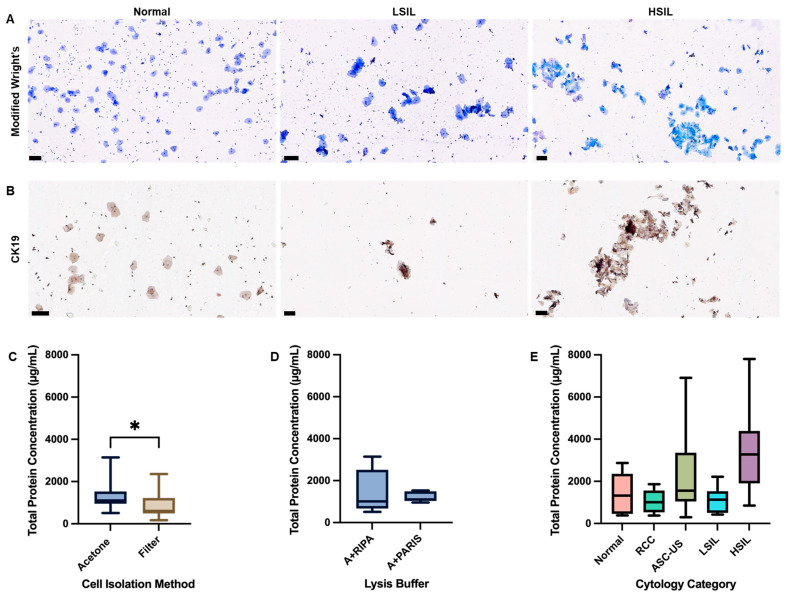
Cell isolation and total protein extraction from clinical cervical swab samples. Representative images of (**A**) modified Wright’s stain, producing blue to purple coloration, with the nuclei stained as dark blue/purple and the cytoplasm appearing as pale blue/purple around the nuclei, and (**B**) ICC staining of the CK19 marker in normal, LSIL, and HSIL swab samples; brown staining demonstrates CK19 expression in the cells (scale bars = 100 μm). (**C**) Total protein yield of acetone precipitation- and filter-based cell isolation methods, independent of the lysis buffer used for protein extraction. (Paired *t*-test: *: *p* < 0.05, *n* = 20.) (**D**) Total protein yield of acetone precipitation (labeled as “A”) followed by cell lysis via RIPA or PARIS™ buffers. (Paired *t*-test: *n* = 10.) (**E**) Total protein yield of cervical cells isolated using acetone precipitation and lysed with RIPA buffer. Cells were from normal (*n* = 14), RCC (*n* = 9), ASC-US (*n* = 33), LSIL (*n* = 12), and HSIL (*n* = 12) swabs. Boxplots in (**C**–**E**) represent the median (bar), interquartile range (IQR, box), and 1.5 × IQR (whiskers).

**Figure 4 cancers-17-01763-f004:**
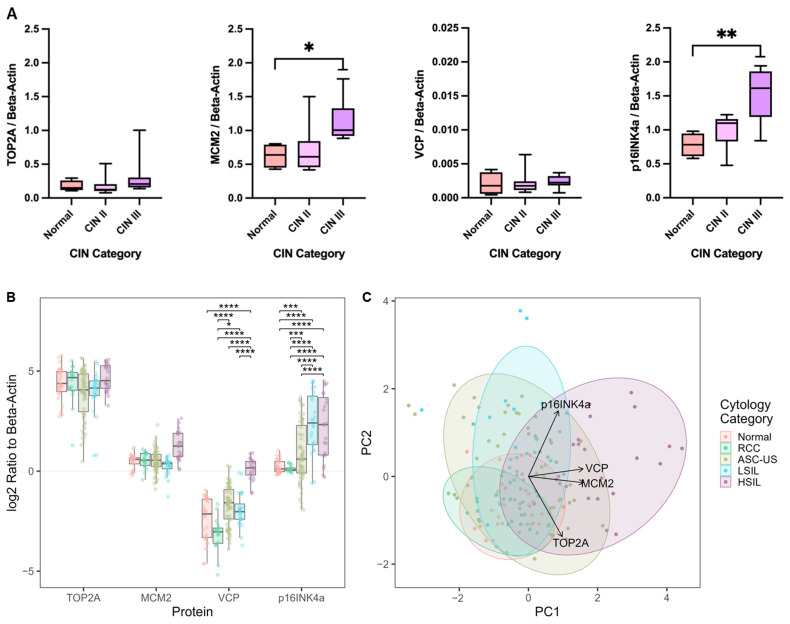
Biomarker expression reported as ratios to beta-actin in lysates of clinical cervical swab samples. (**A**) ELISA measurements of TOP2A, MCM2, VCP, and p16INK4a normalized to beta-actin in lysates of a set of clinical cervical swab specimens with known CIN categories: normal (*n* = 4), CIN II (*n* = 8), and CIN III (*n* = 8). (Dunnett’s multiple comparisons test: *: *p* < 0.05, **: *p* < 0.01.) (**B**) Beta-actin-normalized ELISA measurements of the four target proteins in lysates of clinical cervical swabs with known cytology categories: normal (*n* = 14), RCC (*n* = 9), ASC-US (*n* = 33), LSIL (*n* = 12), and HSIL (*n* = 12). Statistical significance was assessed using a Gamma-distributed log-link GLMM, with fixed effects for proteins, cytology categories, and their interactions, and random effects for biological replicates. (Holm-corrected pairwise *t*-test: *: *p* < 0.05, ***: *p* < 0.001, ****: *p* < 0.0001; the points correspond to individual data points.) (**C**) PCA of biomarker concentrations across different cervical cytology categories. Boxplots in (**A**,**B**) represent the median (bar), interquartile range (IQR, box), and 1.5 × IQR (whiskers).

**Figure 5 cancers-17-01763-f005:**
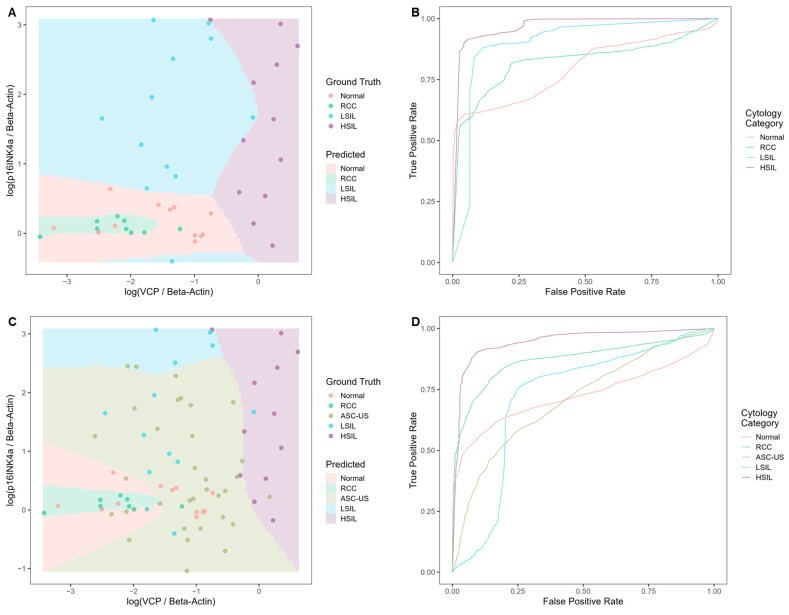
Classification of cervical swab samples’ cytology categories based on the beta-actin-normalized VCP and p16INK4a concentrations. (**A**,**C**) Predicted versus ground truth classifications mapped to biomarker-to-beta-actin ratios. (**B**,**D**) The corresponding ROC curves. (**A**,**B**) demonstrate the results for the classifier fitted to the training data including the normal, RCC, LSIL, and HSIL cases. (**C**,**D**) correspond to the results of the same model fitted to the same training data including the ASC-US cases. The classification was performed using a multinomial GAM. Prediction regions (**A**,**C**) and ROC curves (**B**,**D**) were based on bootstrap cross-validation with 200 replications, using 25% of cases in each cytology class as the test set for each replication. The areas under the curve (AUCs) in (**B**) were 79.9% (normal), 83.2% (RCC), 91.8% (LSIL), and 97.9% (HSIL). For (**D**), the AUCs were 73.1% (normal), 87.5% (RCC), 71.4% (ASC-US), 72.4% (LSIL), and 95.2% (HSIL).

**Figure 6 cancers-17-01763-f006:**
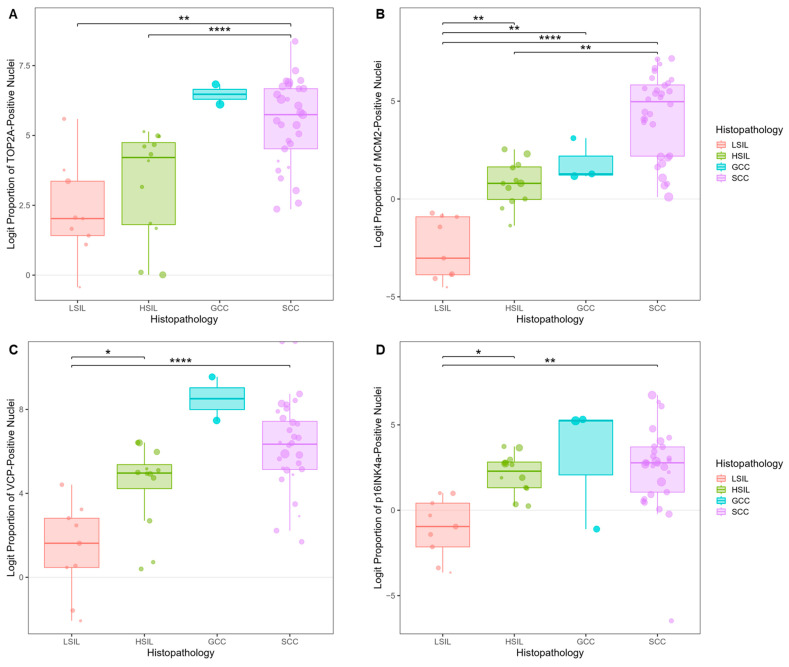
Immunostaining and quantification of biomarker expression in tissues of cervical SILs and CC subtypes. Quantification of (**A**) TOP2A, (**B**) MCM2, (**C**) VCP, and (**D**) p16INK4a in IHC tissue stainings of cervical SILs (LSIL; *n* = 9, and HSIL; *n* = 12) and carcinomas (GCC; *n* = 2–3, and SCC; *n* = 29–32). The quantitative analysis was performed by measuring the proportion of positive nuclei for each biomarker in TMAs of different histopathology diagnoses. Each point represents an individual case, with point size being proportional to the number of nuclei in the core and logit transformation applied to proportions for improved visualizations. Statistical significance was assessed using a logit-link quasibinomial generalized linear model. (Holm-corrected pairwise *t*-test: *: *p* < 0.05, **: *p* < 0.01, ****: *p* < 0.0001. Boxplots represent the median (bar), interquartile range (IQR, box), and 1.5 × IQR (whiskers).) (**E**) Representative images of original IHC-stained sections (scale bars = 200 μm) and their computationally pseudocolor-coded images (scale bars = 100 μm) that were quantified using the Visiopharm software, version 2024.07.2.17285x64. Brown represents the original IHC staining, purple denotes the nuclear counterstain, while pseudocolors indicate staining intensity: red (strong), orange (moderate), and yellow (weak).

**Figure 7 cancers-17-01763-f007:**
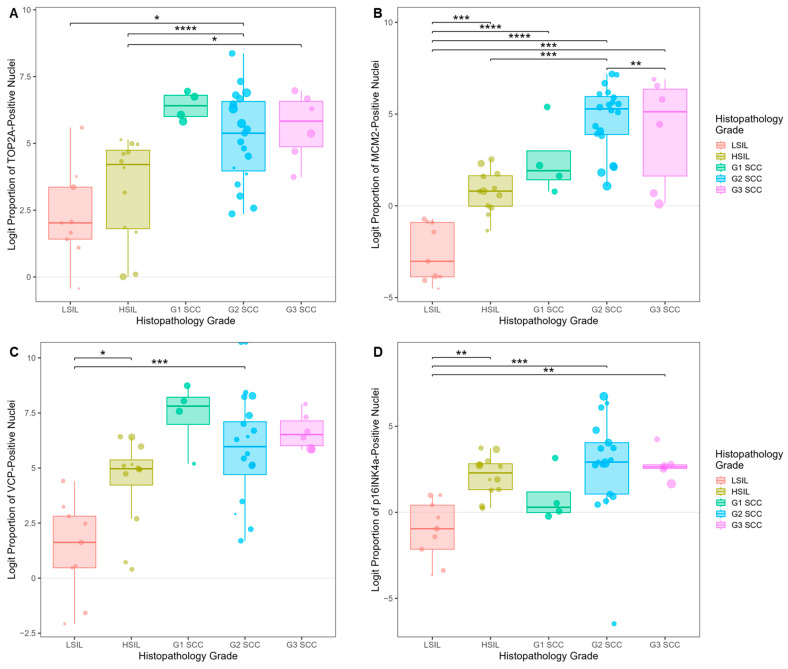
Immunostaining and quantification of biomarker expression in tissues of cervical SILs and SCC grades G1–3. Quantification of (**A**) TOP2A, (**B**) MCM2, (**C**) VCP, and (**D**) p16INK4a in IHC tissue stainings of cervical SILs (LSIL; *n* = 9, and HSIL; *n* = 12) and three SCC grades (G1; *n* = 4, G2; *n* = 17–20, and G3; *n* = 5-6). The quantitative analysis was performed by measuring the proportion of positive nuclei for each biomarker in TMAs with different histopathological grades. Each point represents an individual case; the point size is proportional to the number of nuclei in the core, and proportions were logit-transformed to enhance visibility. Statistical significance was assessed using a logit-link quasibinomial generalized linear model. (Holm-corrected pairwise *t*-test: *: *p* < 0.05, **: *p* < 0.01, ***: *p* < 0.001, ****: *p* < 0.0001. Boxplots represent the median (bar), interquartile range (IQR, box), and 1.5 × IQR (whiskers).) (**E**) Representative images of the IHC-stained sections pseudocolor-coded and quantified using the Visiopharm software, version 2024.07.2.17285x64. Brown represents the original IHC staining, purple denotes the nuclear counterstain, while pseudocolors indicate staining intensity: red (strong), orange (moderate), and yellow (weak) (scale bars = 150 μm).

**Figure 8 cancers-17-01763-f008:**
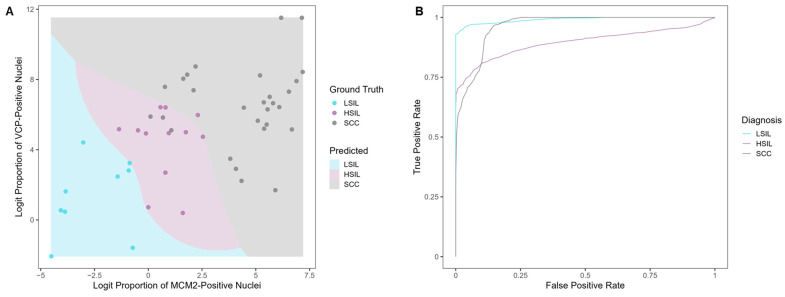
Classification of cervical tissues’ histopathological diagnoses based on the proportion of MCM2- and VCP-positive nuclei. (**A**) Predicted versus ground truth classifications mapped to the proportion of tumor cells expressing MCM2 and VCP. The training data included LSIL, HSIL, and SCC cases. (**B**) The corresponding ROC curves (areas under the curve: LSIL = 98.9%, HSIL = 87.7%, and SCC = 95.6%). Prediction regions (**A**) and ROC curves (**B**) were based on bootstrap cross-validation with 200 replications, using 33.3% of the data in each diagnosis class as the test set for each replication.

## Data Availability

Data are available upon reasonable request.

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
