# Peer review of "Protein Biomarkers Enable Sensitive and Specific Cervical Intraepithelial Neoplasia (CIN) II/III+ Detection: One Step Closer to Universal Cervical Cancer Screening"

_cancers, 2025, doi:10.3390/cancers17111763_

Round 1

Reviewer 1 Report

Comments and Suggestions for Authors

            The authors begin by stating that cervical cancer screening is costly, emphasizing that this issue primarily affects low-income and developing countries. However, they proceed to propose expensive diagnostic strategies that do not offer any tangible reduction in the overall expenditures related to cervical cancer prevention and management.

            Immunohistochemical diagnostics used to predict the progression to cervical cancer are not novel. Such analyses were already being conducted over two decades ago and are now widely implemented in daily clinical practice. This is not about citing my own publications, but rather about maintaining scientific rigor and acknowledging that such challenges have already been undertaken many years ago.

Przybylski Marcin, Promotor: Spaczyński Marek.: Immunohistochemical and serological markers of cervical intraepithelial neoplasia progression. Doctoral dissertation.: Poznań, 2004

Immunohistochemical and serological markers of CIN progression.: 22nd International Papillomavirus Conference and Clinical Workshop 2005. Vancouver, Canada, April 30-May 6, 2005. Book of Abstracts.

Markers of early CIN progresion.: 6th International Multidisciplinary Congress "Human Papillomavirus Infection and Global Prevention of Cervical Cancer. Priorities, Practice and New Directions". Paris, France, 23-26 April 2006. Final Program and Abstracts.

            In my view, a greater contemporary challenge lies in public education rather than financial constraints. In fact, some low-income countries have managed to implement more effective screening programs than wealthier nations, where antivaccine movements and other groups that undermine medical science often have considerable influence.

            I was particularly concerned by the authors’ dismissive stance toward primary prevention of cervical cancer. Recent studies suggest that prophylactic HPV vaccines may also have therapeutic effects. For example: Effect of vaccination against HPV in the HPV-positive patients not covered by primary prevention on the disappearance of infection (Sci. Rep. 2025, Vol. 15, Article 12642, pp. 1–12). While further investigation is certainly warranted, ignoring such emerging evidence reflects a troubling lack of scientific objectivity that serious authors should avoid.

            Despite these concerns, the study itself has been properly designed and adheres to accepted principles of scientific research. The topic remains highly relevant. I support the authors’ efforts toward publication and believe that, following revisions—particularly with regard to the references—the article will represent an innovative and meaningful contribution to the ongoing discussion on cervical cancer prevention.

Author Response

Please see the attachment. Thank you for your time and constructive feedback.

Reviewer 2 Report

Comments and Suggestions for Authors

The article titled " Protein Biomarkers Enable Sensitive and Specific CIN II/III+ Detection: One Step Closer to Universal Cervical Cancer Screening" presents a significant contribution to the literature on cervical cancer.

Cervical cancer is the fourth most common cancer among women globally and disproportionately affects those in low- and middle-income countries (LMICs). Current screening methods are expensive, time-consuming, and often unavailable in resource-limited settings.  In this study, the authors validate critical protein biomarkers, including topoisomerase II alpha (TOP2A), minichromosome maintenance complex component 2 (MCM2), valosin-containing protein (VCP), and cyclin-dependent kinase inhibitor 2A (p16INK4a), for point-of-care cervical cancer screening. By optimizing a simple and efficient protein extraction protocol, the study demonstrates the potential of these biomarkers to detect pre-cancerous lesions and differentiate cervical cancer subtypes. These findings lay the groundwork for the development of low-cost, rapid, and accurate screening assays that could significantly improve early-stage cervical cancer diagnosis and treatment, particularly in LMICs.

Concerns/suggestions:

  1. Sample Size: The relatively small sample size may have contributed to the observed discrepancy in biomarker overexpression between dysplastic cervical tissues and cervical swabs. A larger, more diverse cohort would strengthen the statistical power and generalizability of the findings.
  2. Standardization of Sample Collection: To reduce variability and improve the clinical applicability of these biomarkers, the authors should consider implementing standardized methods for sample collection, handling, and processing. This would enhance the reproducibility of results across different settings.

These findings provide new insights into the quantitative analysis of biomarkers, along with the subsequent sensitive and specific clinical classification, highlights their potential application in SIL early detection and CC prevention, particularly in LMICs. The manuscript is well-articulated and draws reasonable conclusions.

Author Response

(The authors gave the same response as above.)
